# The Challenge of Combining High Yields with Environmentally Friendly Bioproducts: A Review on the Compatibility of Pesticides with Microbial Inoculants

Mariana Sanches Santos [1,2], Thiago Fernandes Rodrigues [1,2], Marco Antonio Nogueira [1] and Mariangela Hungria [1,2,*]





[1] Embrapa Soja, C.P. 231, Londrina 86001-970, Brazil; mari_sanches_s@hotmail.com (M.S.S.); thiagoferrodrigues@hotmail.com (T.F.R.); marco.nogueira@embrapa.br (M.A.N.)

[2] Department of Biochemistry and Biotechnology, Universidade Estadual de Londrina, C.P. 60001, Londrina 86051-990, Brazil

[*] Correspondence: mariangela.hungria@embrapa.br or biotecnologia.solo@hotmail.com; Tel./Fax: +55-4333716206

**Abstract:** Inoculants or biofertilizers aiming to partially or fully replace chemical fertilizers are becoming increasingly important in agriculture, as there is a global perception of the need to increase sustainability. In this review, we discuss some important results of inoculation of a variety of crops with rhizobia and other plant growth-promoting bacteria (PGPB). Important improvements in the quality of the inoculants and on the release of new strains and formulations have been achieved. However, agriculture will continue to demand chemical pesticides, and their low compatibility with inoculants, especially when applied to seeds, represents a major limitation to the success of inoculation. The differences in the compatibility between pesticides and inoculants depend on their active principle, formulation, time of application, and period of contact with living microorganisms; however, in general they have a high impact on cell survival and metabolism, affecting the microbial contribution to plant growth. New strategies to solve the incompatibility between pesticides and inoculants are needed, as those that have been proposed to date are still very modest in terms of demand.

**Keywords:** rhizobia; plant-growth-promoting bacteria; fungicide; insecticide; herbicide; biological nitrogen fixation; inoculation

## 1. Introduction

Technologies and agricultural inputs currently applied for food production are essential for large-scale production and are mandatory to feed a population of more than 7 billion people [1]. Years of research and experiments continually performed into the challenges and technological evolution have resulted in progress in several fields of science. The 1950s was known as the "Green Revolution" period, marked by the intense modernization of agriculture [2,3]. Products including new machines and synthetic fertilizers, with an emphasis on nitrogen (N) fertilizers, pesticides, seeds of better quality, improvements in the water supply systems, breeding and genetic engineering are examples of technologies developed at that time and that have gained prominence in agriculture [4,5].

The main positive result of the Green Revolution was the global increase in food production, thus contributing to the reduction of hunger in the world. However, in the following years, the side-effect of this revolution unfolded [6]. The accumulation of pesticides and chemical fertilizers contributes to the pollution of groundwater and cultivated land, soil degradation, and reduction of biodiversity in different ecosystems [5,7]. Increased deforestation, soil degradation, and emission of polluting gases into the atmosphere have been increasingly observed [8–10]. Despite the undeniable benefits of the Green Revolution, many of the technologies and inputs generated during that period are broadly criticized today.

Currently, efforts are being made towards the development of new technologies and inputs focused on more sustainable systems.

Contemporary scientists have pointed out that we are living in a "New Green Revolution" whose main characteristic, and which differs from that experienced in the 1950s, is the development of environmentally-friendly technologies and products [3,11–13]. Examples of this new concept include the development of products and techniques such as crop rotation, plant genetic engineering for resistance to pests, diseases, and abiotic stresses such as drought, the use of bio-inputs as activators of soil biota, biopesticides and microbial inoculants, also known as biofertilizers in some countries, with the purpose of partially or fully replacing the use of chemical fertilizers, favoring the growth of plants [14–17].

Although the current movement towards agricultural sustainability has force worldwide, the use of agrochemicals is and will continue to be the reality of most farmers [18]. As such, the common scenario towards improving agricultural sustainability with feasible yields to guarantee food security includes the increasing use of bioproducts, such as microbial inoculants, together with pesticides, which are still indispensable for controlling pests and diseases [4,5,11,14,17]. Therefore, the compatibility between inoculants and pesticides must be understood.

In general, pesticides contain molecules that are potentially toxic to living cells. Depending on the specificity, pesticides can cause toxicity to cells of microorganisms, animals, and plants, often resulting in death after contact with the product. In agriculture, they are commonly applied to the seeds, soil, and leaves of plants to prevent or control pests and diseases [7,18–31]. Usually, at least for large commercial crops such as is the case of soybean [*Glycine max* (L.) Merr.] [11,15,17,20] and maize (*Zea mays* L.) [26–28] in South America, pesticides and inoculants are added together on the seeds. Thus, it is necessary to verify whether microbial cells in the inoculants are affected by pesticides, impairing the benefits of inoculation.

We should also mention the increasing demand for anticipated inoculation or pre-inoculation [20], and in the pre-inoculation the most common adoption is to treat seeds with pesticides and inoculants several days before sowing [15,24]. However, the microorganisms are subjected to long-term exposure to pesticides, increasing the pernicious effect on the bacteria and resulting, for example, in decreased nodulation in legumes [21–23], lower N accumulation in grains [24,25], and negatively impacting root development of grasses [26]. These losses may be due to microbial cell death caused by pesticides, as demonstrated in some studies [25–28], in which the longer the contact between bacteria and pesticides, the greater the mortality. Moreover, changes in cell metabolism, such as formation of smaller colonies and decreased nitrogenase activity, have been reported [25].

The world will need more food, and to meet this increased demand, pesticides should continue to be required, at least in a medium term period. However, there is also an increasing demand for environmentally-friendly inputs, including inoculants, to replace chemical fertilizers, and biopesticides to replace chemical pesticides. One major challenge is to make the chemicals and biologicals compatible. In this review, we gathered information on the use of pesticides and inoculants, starting with the history, current situation, and results of studies that investigated the effects of fungicides, insecticides, and herbicides on microbial inoculants.

## 2. History and Use of Pesticides

Studying planting and cultivation practices in ancient times, historians have reported that civilizations were already searching for effective approaches to protect and preserve their food. For millennia, methods such as burning sulfur, using arsenic, growing toxic species together with the crop of interest, and using salts and ashes against weeds were used to protect crops [29,30]. In one of the oldest documents, from approximately 1550 BC, called the Ebers Papyrus, interesting information about techniques used to eliminate insects from food planting areas has been described [30]. The report describes a mix of mercury and arsenic that was used for pest control [30], and a century later, arsenic was used along

with honey, especially against ants. In 1867, during a potato beetle [*Leptinotarsa decemlineata (Say)*] outbreak in Colorado, USA, arsenic was used for pest control [29]. During the 1850s, a vineyard owner from Bordeaux, France, applied a mixture of copper and lime to grapes. Initially, the objective was to keep thieves away from his vineyards, but the wine producer realized that the application resulted in lower incidence of diseases. Notably, this mixture is still used today as a fungicide. Over the years, insecticides derived from plants have been discovered, such as pyrethrins and nicotine, the latter used specially to control aphids [19].

In 1939, Paul Müller discovered dichlorodiphenyltrichloroethane (DDT), which was the first modern pesticide. This product played an important role during World War II when it was broadly used to control diseases transmitted by insects, such as malaria and typhus, ensuring soldiers' health. The discovery of DDT resulted in considerable benefits to agriculture and human health, resulting in Müller being awarded the Nobel Prize in Medicine in 1948. Numerous other cheap and effective synthetic organic pesticides have been developed, contributing to a breakthrough in the market and starting a new era in pest and disease control [19,29–31]. Fungicides such as captan and glyodin, the insecticide malathion, and the herbicide triazine were introduced in the following decades [29].

The use of pesticides increased until 1962 when the development of new products began to slow down because of the first studies and reports on the environmental and health risks associated with the indiscriminate use of pesticides. The book "Silent Spring" (1962), authored by the American scientist Rachel Carson, played an important role in the history of pesticides. In the book, the author discusses the harmful effects caused by the field spraying of several pesticides containing chlorinated hydrocarbons, among them the most important at the time, DDT. The effectiveness of these products is closely related to their stability and persistence in the environment, but they are also able to accumulate in the adipose tissue of some animals, a process known as bioaccumulation, which in some cases results in biomagnification, factors that make these compounds highly dangerous [19,29,30,32]. Another important finding was the confirmation of the housefly (*Musca domestica*, L) resistance to DDT in Sweden only after a few years of application, another negative point for the use of this product [19].

As a result, in 1972, the US Environmental Protection Agency (EPA) banned the use of DDT in the country, and several pesticides were classified as restricted use, for example, endosulfan, dieldrin, and lindane. Organophosphorus and carbamates, which have lower risk, had been suggested as alternatives [19,29]. DDT was banned in several other countries and, in 2001; during the Stockholm Convention, 179 nations signed a treaty that agreed to ban 12 persistent organic pollutants (POPs). It is interesting to note that since the 1960s, when both pesticides production and use became strictly regulated, alternative methods for pest and disease control began to be studied, with an emphasis on biological control (BC) [19,29,30] and integrated pest management (IPM) [33]. The principle of IPM is based on understanding population dynamics and using actions compatible with the environment to minimize the incidence of pests. In 1998, Kogan defined IPM as "the intelligent choice and use of control tactics that will produce favorable consequences from an economic, ecological, and sociological point of view." In conclusion, the principle is to use several compatible techniques to keep the pest population at levels below the capacity that causes economic, social, and environmental damage. IPM is currently widely used in crops around the world and is responsible for excellent results like reduced use of chemicals and increase in yields, in addition to other traditional methods such as mechanical and physical control and use of resistant plants, but is not able to fully replace the use of chemical pesticides, which are still used on a large scale [34].

In 1990, the average worldwide use of pesticides by cultivation area was of 1.5 kg ha$^{-1}$. Almost 30 years later, this value grew considerably, reaching an average of 2.63 kg ha$^{-1}$ in 2018. The continents that applied more pesticides in 2018 were Asia and America, reaching 3.67 kg ha$^{-1}$ and 3.52 kg ha$^{-1}$, respectively. Europe applied 1.66 kg ha$^{-1}$ of pesticides, while in Africa the average application was 0.3 kg ha$^{-1}$ [18]. Among Asian countries, China, Japan, and Korea had the highest averages of pesticides per hectare in 2018, reaching 13.07 kg ha$^{-1}$,

11.84 kg ha$^{-1}$, and 11.73 kg ha$^{-1}$ respectively. High rates were also reported in 2018 for South American countries, including Ecuador (25.8 kg ha$^{-1}$), Uruguay (8.16 kg ha$^{-1}$), Brazil (5.94 kg ha$^{-1}$), Chile (5.86 kg ha$^{-1}$), and Argentina (4.29 kg ha$^{-1}$), while the United States of America (USA) applied an average of 2.54 kg ha$^{-1}$ of pesticides in the same year [18].

Each country has its own laws and regulations regarding the production, commercialization, and use of pesticides, designed mainly to protect human and environmental health. Among the actions of regulatory agencies are, for example, limitation of species to which a certain pesticide can be used, the requirement to use protective equipment, and the total prohibition of any product that is proven to be dangerous, which cannot be reliably mitigated [35]. Estimates point out that since 1970, 508 types of active ingredient have been used in the USA, 134 of which were banned, with the majority of the cancelations taking place voluntarily by the manufacturers; only 37 prohibitions came from judicial decisions. The United States is behind other countries with regard to banning pesticides, probably due to deficiencies in the legislation. From the list of pesticides still in use in the United States, 72 of them have been banned in the European Union, 17 in Brazil, and 11 in China [35].

Several studies have been carried out in the past few decades to understand the damage to human health and the environment caused by certain chemical pesticides [36–38]. These studies are very important because they generate information to guarantee food security. However, crop management still requires the application of pesticides to achieve high yields to meet the world´s increasing demand for food. Given this scenario, the indications are of a continuing use of pesticides, but with a trend toward more environmentally friendly formulations, such as the replacement by biological control.

## 3. Inoculants or Biofertilizers

Following the trend of agricultural production and concern about environmental sustainability, an innovative biotechnological product based on living microorganisms capable of making nitrogen available to plants was patented in 1896 and launched in 1898 by the first inoculant producing company, Nitragin, replacing the application of potentially polluting N fertilizers. The first inoculant contained nitrogen-fixing rhizobia for soybean crop [17,39–41]. Rhizobia are diazotrophic bacteria with an enzymatic apparatus to realize the biological nitrogen fixation (BNF) process, in which atmospheric nitrogen ($N_2$) is converted into ammonia ($NH_3$) and further to organic compounds that are easily assimilated by plants (Figure 1). Therefore, when diazotrophic microorganisms are associated with specific plants, they supply nitrogen to their host, contributing to their development, and in return receive photosynthates from the host plant for their metabolism [42,43].

Since the first inoculant was launched, a variety of inoculants has been produced, including rhizobia and other plant growth-promoting bacteria (PGPB) [17]. It has been necessary to research on several fronts, including the selection of microorganisms for each plant species [44–49], development of the culture media [50,51], search for inoculant carriers [52–55], development of large-scale production, product distribution logistics, methods of inoculation [56–58], among others. Such studies have been responsible for expanding, diversifying, and improving the quality of inoculants. There are several reports on the contribution of inoculants increasing yields of crops at a low cost and mitigating the environmental impact [15,17].

As the production and commercialization of inoculants have increased, some countries have created laws to standardize, supervise, and guarantee the safety and quality of these bioproducts. In 1954, a microbiology professor at the University of Sydney, Australia, listed basic recommendations for quality control and use of legume inoculants, establishing the first quality control laboratory in the country. The Australian Inoculant Research Group (AIRG) is responsible for quality control. Since 2010, Australian inoculants that meet quality standards have exhibited a trademark called "Green Tick Logo," which certifies that the product contains the correct strain, the number of living cells equal to or above the standard, and the minimum number of contaminating microorganisms [59]. Australian legislation served as a basis for legislation in other countries, such as in Uruguay and Brazil.

In Brazil, legislation was established in 2004 and updated in 2011 [60–62]. Australia has moved to voluntary participation in the quality control of commercial inoculants, while in other countries such as France and Canada, as well as in many South American countries, participation is mandatory. The third group, including the US, comprises only internal control by the industry [17,63].

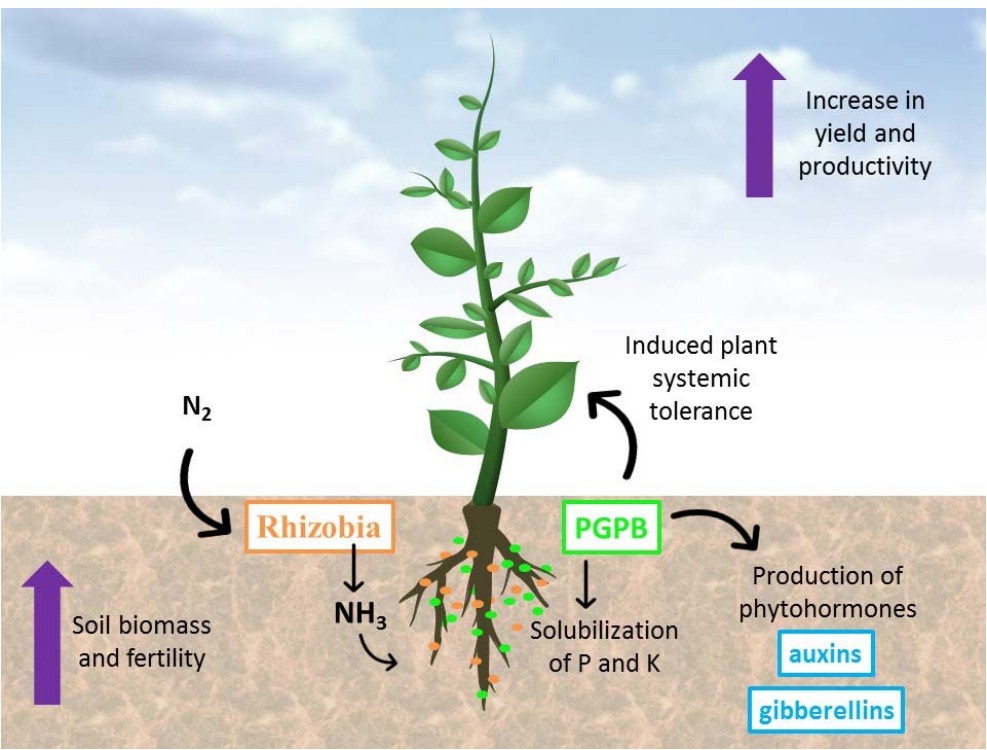

**Figure 1.** Main benefits of inoculation with rhizobia and plant-growth-promoting bacteria (PGPB). Biological nitrogen fixation (BNF), phosphate (P) and potassium (K) solubilization, phytohormone production (auxins and gibberellins), and induced plant systemic tolerance to abiotic and biotic stresses are represented. As benefits, there are increases in biomass production, yield, and improvement in soil fertility.

Brazilian standards include a list of bacterial strains authorized for inoculation in different plant species, establishing a minimum cell concentration, and the limits for contaminants. Inoculants containing rhizobial species must present $1 \times 10^9$ CFU (colony-forming units) per gram or mL until the expiration date, which must be at least 6 months, but other species may register with lower concentrations [61,62]. The maintenance of cell concentration contributes to achieving the desired performance and ensuring product quality, but it is a challenge for the industry, as several factors can affect cell survival such as temperature, pH, drought, light, and availability of nutrients [50,51,53,63,64].

Soybean inoculation is certainly the most successful example worldwide, with an emphasis in South America. For example, in Brazil, it is well known that inoculation with elite *Bradyrhizobium* spp. strains fully supplies the plant's demand for nitrogen, avoiding the use of N-fertilizers even with high-yielding genotypes [15,42]. In the 2019/20 crop season, approximately 36 million hectares were cropped with soybean in Brazil, and even though most of the area had been inoculated in previous seasons, about 80% of the farmers adopted annual inoculation [65,66]. Traditionally, soybean in Brazil has been inoculated only with *Bradyrhizobium* spp. strains [15]; however, in the past five years, co-inoculation of *Bradyrhizobium* spp. with the PGPB *Azospirillum brasilense* has been increasingly used, so that in a short period it has been adopted by 25% of the farmers [65,66]. The main trait of the *A. brasilense* strains used for co-inoculation of soybean in Brazil has been recognized as the synthesis of phytohormones [67,68].

Besides phytohormones synthesis, beneficial properties associated with PGPB include BNF, phosphate and potassium solubilization, production of siderophores, detoxification of heavy metals, induction of plant systemic tolerance to abiotic and biotic stresses, production of hydrolytic enzymes, and production of exopolysaccharides [69–75] (Figure 1). Such properties have been reported in several microorganisms, and the most commonly cited carrying one or more of these properties are *Azospirillum* spp. [72,76], *Pseudomonas* spp. [77–79], and *Bacillus* spp. [80,81], among others [71,82]. With important properties and taking advantage of different microbial processes, the inoculation with mixes of bacteria has gained increased attention [67,68,82–84]. Some proposed mixes combine several species [75,82], but one knows that it is difficult to grow and maintain proper concentrations of bacteria with different metabolic needs, In addition, a thorough preliminary study must be carried out so that the chosen species are compatible with each other.

Before being used as a co-inoculant for legumes, PGPB such as *A. brasilense* has long been used in the inoculation of non-legumes, especially grasses such as maize, wheat (*Triticum aestivum* L.), and rice (*Oryza sativa*). Inoculants containing *A. brasilense* have been commercialized for more than 20 years; since 1996 in Argentina [76], 2002 in Mexico [85], and 2009 in Brazil [17,44,66]. *A. brasilense* can fix nitrogen, but in much lower quantities than rhizobia, making necessary supplemental N-fertilizer. The possibility of reduction of approximately 25% of the N-fertilized has been observed when *A. brasilense* is used as an inoculant in maize and this is environmentally and economically important [44,57,66,86]. One of the greatest benefits of *A. brasilense* is its ability to produce phytohormones, such as auxins, that stimulate root development [72,87], and gibberellins [87]. In addition, *A. brasilense* can induce plant defense mechanisms under abiotic and biotic stresses [69,70,72,73]. Due to excellent results of inoculation with *A. brasilense* in several crops [88], it is expected that this practice will continue to grow in the coming years. In Brazil, in a short period of 10 years, since the first commercial inoculant containing *A. brasilense* was launched, the number of commercialized doses reached 10.5 million in 2020 [66].

Rhizobial inoculants have been traditionally used in pasture-growing legumes, mainly alfalfa (*Medica sativa* L.) [89]. However, as most pastures worldwide have grasses, the use of other PGPB in important pastures such as brachiarias (*Urochloa* spp.) has increasingly attracted attention, with results showing improvement in soil fertility, biomass yield, and nutrient content in the forages (Figure 1) [12,90–92]. The inoculation pastures of grasses with PGPB has environmental and economic importance because most of the pastures worldwide are at some stage of degradation [93,94]. By offering better quality fodder to cattle, the pastures can hold a larger number of animals, making saving of other areas from transformation into pastures.

There are different ways to deliver the inoculants to the crops. Seed inoculation is the most common and easy method, as it can be carried out using solid or liquid inoculants. In this process, the inoculant is applied to the surface of the seeds, with or without stickers, which are then shaken so that the product spreads evenly. Other methods of applying liquid inoculants are possible, such as spraying on the soil surface or applying the product in the sowing furrow. Another alternative is leaf spraying, which can be done during certain periods of plant growth. Regardless of the strategy of inoculation, it is extremely important to apply the correct dose of the product, according to the manufacturer's recommendations, as lower or higher cell concentrations may decrease inoculation efficiency [56,57,67,95].

## 4. Are Pesticides and Microbial Inoculants Compatible?

As the benefits of inoculation are closely related to the establishment of a plant-microorganism interaction, the survival and maintenance of microbial properties are crucial and mandatory. Therefore, the evaluation of microbial survival at the time of inoculation, and of the effectiveness on the inoculated crop are critical. The most common situation in commercial crops is the combination of several products for different purposes, such as soybean seed treatment with microbial inoculants for nitrogen fixation, fertilizers and pesticides for the prevention or treatment of pests and diseases. In many cases, depending

on the mode of application, one product ends up being exposed to the other, or interacting with one another either on the seeds, propagated material, in the soil, or leaf surface. It is important to know whether the contact of pesticides with microorganisms in the inoculant can affect cell survival and metabolism. Concerns about the compatibility of agrochemicals with microbial inoculants have been raised for decades, and several studies have shown that the impact of chemicals on the inoculant depends on the active ingredient, the presence of other chemical substances that make up the formulation of pesticides, the mechanism of action (systemic or contact), and the method of application. The effects of incompatibility also depend on the bacterial species present in the inoculant, which may have different responses. However, few species have been evaluated for this purpose. The most recurrent ones belong to the genres *Rhizobium* spp., *Bradyrhizobium* spp. and *Azospirillum* sp.

### 4.1. Compatibility with Fungicides

Fungicides are chemical products formulated to prevent the infection of plant tissues by phytopathogenic fungi, and in some cases, capable of extending the control of diseases caused by bacteria and viruses. The control exerted by fungicides can be mediated by killing the pathogen, temporarily inhibiting its germination and growth, or by affecting the production of spores [96]. Over the years, several fungicides have made it to the market; some have stood out and remained at the top in the list of the most used fungicides until today, more effective products have replaced others, and some have been banned. Fungicides of contact generally do not have a specific mode of action, are highly toxic, and when applied to seeds, soil, and plant leaves limit the pathogen survival. The most common are thiram (dithiocarbamate), captan (quinone and heterocyclic), exon (aromatic), and guazatine. Upon entering microbial cells, the molecules promote a series of chemical reactions in nucleic acids and their precursors, and metabolic routes, affecting cell survival [96].

The majority of studies on compatibility have been performed with fungicides and microbial inoculants carrying rhizobia. Fungicides may affect several steps of the symbiosis, from the survival of the rhizobia on the seed to nodule formation and $N_2$ fixation efficiency; in general, studies have been performed with soybean (e.g., [53,97–99]). The use of pesticides intensified in the past two decades, and so did concerns about their compatibility with inoculants [17].

Brikol et al. [21] evaluated the effects of applying different concentrations of the fungicide Thiram (10 to 750 $\mu$g mL$^{-1}$) on soybean seeds inoculated with *B. japonicum*, which were grown under greenhouse conditions for 75 days. They observed that concentrations above 100 $\mu$g mL$^{-1}$ reduced nodule number and dry weight, as well as the activity of the bacterial enzyme nitrogenase, responsible for the nitrogen fixation process. Similarly, there are reports [24,100] of decrease in soybean nodulation and N accumulation in plants when seeds were inoculated with *B. elkanii* (SEMIA 5019) and *B. japonicum* (SEMIA 5079) and treated with different fungicides, benomyl, captan, carbendazin, carboxin, difenoconazole, thiabendazole, thiram, and tolylfluanid. Changes in nodule number were also observed in a field trial by Zilli et al. [101] when soybean seeds were treated with either carbendazin + thiram or carboxin + thiram.

Interestingly, results of some studies indicate differences between strains in their tolerance to fungicides. For example, in soybean seeds treated with carbendazim + thiram, the lowest tolerance was observed in *B. elkanii* amongst four soybean *Bradyrhizobium* strains used in commercial inoculants in Brazil [*B. elkanii* (SEMIA 5019 and SEMIA 587), *B. japonicum* (SEMIA 5079), and *B. diazoefficiens* (SEMIA 5080)] [101]. In another study, Gomes et al. [102] observed no effects on nodulation when seeds were inoculated with *B. japonicum* SEMIA 5079 + *B. diazoefficiens* SEMIA 5080 and treated with carbendazin + thiram. More recently, when the compatibility of *B. japonicum* SEMIA 5079 and *B. elkanii* SEMIA 587 was verified with Standak$^{\circledR}$ Top, composed of a mixture of two fungicides and one insecticide (piraclostrobin, thiophanatemethyl, and fipronil), the effects were also more drastic for *B. elkanii* [25]. Altogether, indications are that *B. elkanii* is less tolerant to fungicides than *B. japonicum* or *B. diazoefficiens*.

It is reasonable to postulate that the main effects of the fungicides used in seed treatment would be the decrease of rhizobial survival or inhibition of the root infection process, consequently affecting nodulation and BNF, and as a result grain yield, as observed by Zilli et al. [101]. In that study, the grain yield reduced by 20%, in addition to a decrease in the N content in grains when seeds were treated with *B. elkanii* SEMIA 587 + carbendazin + thiram (Figure 2). However, some reports have indicated that the effects of fungicides may appear later. For example, in a study by Gomes et al. [102], although fungicides (carbendazin + thiram) did not affect nodulation, plants had lower number of pods per plant, grains per plant, and yield. Intriguingly, in two field experiments performed with soybean in Brazil, seed treatment with Standak® Top affected the total N accumulated in the grains of plants relying on both BNF and N fertilizer, indicating the negative impact of the pesticide on N metabolism [25]. In another study, a decrease in both protein and oleic acid contents was observed in soybean inoculated and treated with mefenoxan + fludioxonil [103].

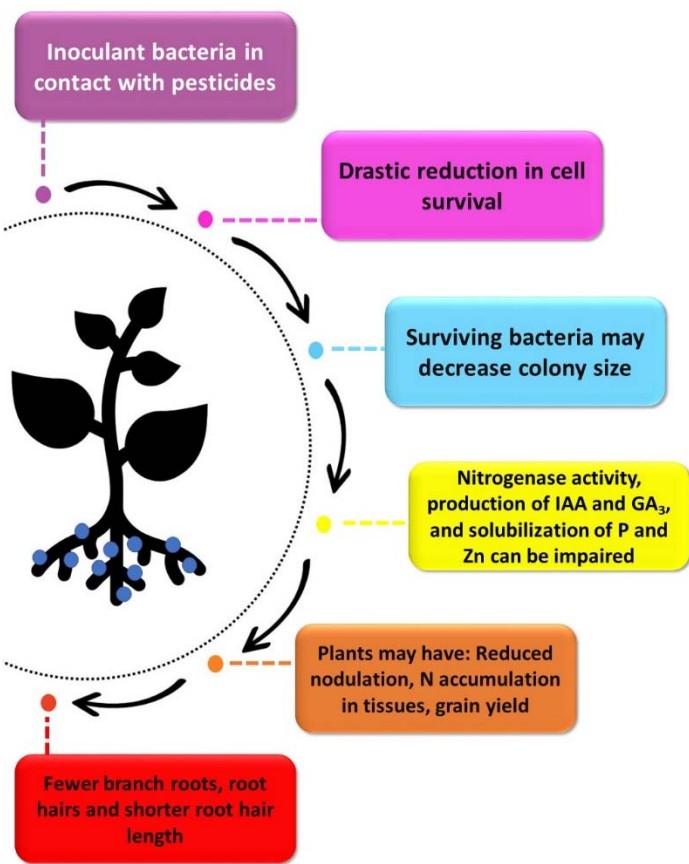

**Figure 2.** Representation of the effects reported on the incompatibility between pesticides and inoculants, from the moment of contact with microbial cells to the damage to plant development.

As mentioned previously, negative effects may be related to the toxicity of fungicides on microbial cells, followed by impacts on microbial metabolism, reducing the effectiveness of the inoculant. Ahmed et al. [104] evaluated the growth of *Bradyrhizobium* sp. and *Rhizobium* sp. in Petri dishes containing solid culture medium and different concentrations of fungicides (captan, thiram, luxan, milcurb, and frernasan-D), soaked in discs placed on the medium. The bacteria were tolerant to concentrations below 100 μg L$^{-1}$, but at 1000 μg L$^{-1}$ inhibition of growth and a decrease in the colony diameter of the surviving cells were observed (Figure 2). Rathjen et al. [105] also reported that *Rhizobium leguminosarum* bv. viciae (WSM1455) was sensitive to the fungicides Thiram 600 and P-Pickel T (PPT), and their active ingredients (thiram and thiabendazole) at concentrations above 200 μg disc$^{-1}$, with growth halos greater than 10 mm around the disks containing the fungicide.

To verify the survival of *B. elkanii* (SEMIA 5019) and *B. japonicum* (SEMIA 5079) on seeds treated with fungicides, Campo et al. [24] recovered and counted living cells from seeds inoculated and treated with benomyl, captan, carbendazin, carboxin, difenoconazole, thiabendazole, thiram, or tolylfluanid. After only 2 h of contact with carbendazin + thiram, viable bacterial cells were reduced by up to 64%, and reached 83% after 24 h (Table 1). Mortality was verified with all other fungicides, reaching 95% with the mixture of thiabendazole and tolylfluanid after 24 h of contact. In addition to rhizobia, fungicides can also impair the contribution of other PGPB. The toxic effect of the combination of carbendazin + thiram was also verified for *A. brasilense* strains Ab-V5 and Ab-V6 by Santos et al. [27]. In that study, a decrease in the recovery of viable cells from seeds only after 2 h of contact was observed, and the viable cell count drastically decreased after 24 h, in comparison with the seeds not treated with fungicides (Figure 2).

**Table 1.** Effect of pesticides on cell viability and/or morphology and on the metabolism of microorganisms used as inoculants.

| Pesticide | Concentration | Microorganism | Effect | Reference |
|---|---|---|---|---|
| Monocrotophos [i], Malathion [i], Chlorpyripho [i], Dichlorvos [i], Lindano [i] e Endosulphan [i] | Recommended dose of each product | *Gluconacetobacter diazotrophicus* | With the exception of Malathion, all insecticides reduced cell viability. Nitrogenase activity was totally inhibited by Monocrotophos, Dichlorvos and Lindano. Production of IAA and GA$_3$, and solubilization of P and Zn were impaired | [106] |
| Butachlor [h], Alachlor [h] Atrazine [h] and 2,4-D [h] | Recommended dose of each product | | Cell growth was hindered by 2,4-D. All herbicides reduced the activity of nitrogenase, the production of IAA and GA$_3$, and the solubilization of P and Zn | |
| Captan [f], Thiram [f], Luxan [f] and Fernasan-D [f] | 1000 µg L$^{-1}$ | *Bradyrhizobium* sp. and *Rhizobium* sp. | Decreased colony diameter and inhibited growth in areas close to the fungicide application site | [104] |
| Benomyl [f], Captan [f], Carbendazin [f], Carboxin [f], Difenoconazole [f], Thiabendazole [f], Thiram [f], Tolylfluanid [f] | Recommended dose for soybean | *B. elkanii* and *B. japonicum* | All fungicides caused mortality of microorganisms | [24] |
| Imidacloprid [i], Fipronil [i], Thiamethoxam [i], Endosulphan [i] e Carbofuran [i] | 250 g ha$^{-1}$, 400 g ha$^{-1}$, 480 g ha$^{-1}$, 2.800 g ha$^{-1}$, 1.650 g ha$^{-1}$, respectively | *Herbaspirillum seropedicae* | Endosulphan increased the lag phase. Carbofuran increased generation time and reduced lag phase | [107] |
| Pyraclostrobin [f], thiophanato-methyl [f] e fipronil [i] | 2 mL kg$^{-1}$ maize seed | *A. brasilense* (strains Ab-V5 and Ab-V6) | Drastic reduction in cell concentration 24 h after inoculation in treated seeds | [26] |
| Carbendazin [f] + Thiram [f] | 40–60 mL 20 kg$^{-1}$ maize seed | | Drastic reduction in cell concentration 24 h after inoculation in treated seeds | [27] |
| metalaxil-m + fludioxonil + tiametoxame + abamectin | Recommended dose for maize | | Drastic reduction in cell concentration 12 h after inoculation in treated seeds | [28] |

**Table 1.** *Cont.*

| Pesticide | Concentration | Microorganism | Effect | Reference |
|---|---|---|---|---|
| Thiram [f], Thiram + Thiabendazole [f] and PPT [f] | >200 µg L$^{-1}$ | *Rhizobium leguminosarum* bv. viciae | Formation of growth inhibition halos greater than 10 mm around the fungicide | [105] |
| Imidacloprid [i] | 0, 100, 200 e 300 µg L$^{-1}$ | | Formation of growth inhibition halos greater than 10 mm around the insecticide for all concentrations evaluated | |
| Pyraclostrobin [f], thiophanato-methyl [f] e fipronil [i] | Recommended dose for soybean | *B. elkanii* and *B. japonicum* | Drastic decrease in cell concentration after 7 days of exposure. Colony formation with smaller diameter | Rodrigues et al. [25] |

[f] fungicide; [i] insecticide, [h] herbicide.

### 4.2. Compatibility with Insecticides

Many herbivorous insects feed on plants during their larval and adult stages, and/or some are important vectors of plant diseases. In both cases, insects may cause serious damages to crops. Insecticides, usually synthetic chemicals, acting as ovicidal, larvicidal, and adulticidal agents are used to prevent growth or kill insects [108]. Neonicotinoids, organophosphates, diamides, pyrethroids, and carbamates act on nerves and muscles; phosphides, cyanides, and carboxamides on respiration, and cyclic ketoenols and ecdysone are agonists that interfere with insect growth and development [109].

Rathjen et al. [105] evaluated the in vitro toxicity of an imidacloprid-based insecticide on *R. leguminosarum* bv. viciae (WSM1455), and *Mesorhizobium ciceri* (CC1192). The strains were applied on solid culture medium in Petri dishes and sterile filter discs containing different concentrations (0, 100, 200, and 300 µg discs$^{-1}$) of the insecticide. The inhibition of *R. leguminosarum* growth was observed at all concentrations. In alfalfa (*Medicago sativa* L.), Fox et al. [22] reported that seed treatment with the insecticides methyl parathion, DDT and pentachlorophenol affected the symbiosis with *Ensifer* (syn. *Sinorhizobium*) *meliloti*. The insecticides not only reduced nodule number and dry weight but also nitrogenase activity in nodules and plant biomass production. Ahemad [23] also reported several negative effects with the application of pyriproxifene, at the recommended dose of 1300 µg kg$^{-1}$ soil, in chickpeas (*Cicer arietinum* L.), peas (*Pisum sativum* L.), mung beans (*Vigna radiata* L. Wiclzek), and lentils (*Lens esculentus, = Lens culinaris* Medik) grown in pots that remained in an open field. Although the plants had not been inoculated, inferring that nodulation was related to indigenous rhizobia, pyriproxifer resulted in a 44% decrease in nodule number in peas, 14% in mung beans, and 5% in chickpeas and lentils, resulting in decreases in nodule dry weight compared with the controls not treated with insecticide. There was also a 17% decrease in the concentration of nitrogen in the roots of chickpeas, 15% in peas, 14% in mung beans, and 18% in lentils, and a 5% decrease in the protein contents in grains of chickpeas, 4% in mung beans, 3% in lentils, and 1% in peas, compared with the controls not treated with insecticide.

Insecticides also affect PGPB other than rhizobia. For example, Fernandes et al. [107] studied the effects of five insecticides (imidacloprid, fipronil, fenamethoxam, endosulfan, and carbofuran) indicated for sugarcane (*Saccharum* spp.) on the diazotrophic bacterium *Herbaspirillum seropedicae*. In vitro evaluations of cell growth after 33 h indicated that the insecticides that most interfered with bacterial growth were endosulfan and carbofuran. Madhaiyan et al. [106] evaluated the effect of different insecticides (monocrotophos, malathion, chlorpyriphos, diclorvos, lindane and endosulfan) on *Gluconacetobacter diazotrophicus*, another PGPB found in association with sugarcane. Except for malathion, all other insecticides reduced cell concentration, and lindane lysed the cells (Figure 2). In the same study, nitrogenase activity was fully inhibited by monocrotophos, dichlorvos, and lindane, 83.3% by chlorpyriphos, 80.9%

by melation, and 33.4% by endosulfan. Concerning the synthesis of indoleacetic acid (IAA) and gibberellin A3 (GA3) by *G. diazotrophicus*, inhibition was observed with lindane, with decreases of 93.2% and 96.5% for IAA and GA$_3$, respectively. The authors also described that the insecticides dichlorvos, chlorpyriphos, and lindane completely inhibited the solubilization of phosphate (P) and zinc (Zn) [106] (Figure 2).

### 4.3. Compatibility with Herbicides

Another class of important pesticides for agriculture are herbicides used for weed control. Herbicides have different degrees of specificity based on differences in biochemical pathways in certain plant groups. The mode of action of herbicides has specific degrees of toxicity, depending on the biochemical differences of the plants, and is generally related to the cell division process [110]. Souza and Guedes [110] gathered studies on the action of several herbicides on *Allium cepa* and *Vicia faba* plants used as bioindicators, and described how most of these studies reported that herbicides induced both a decrease in the mitotic index, and the occurrence of mitotic chromosomal aberrations.

Examples of herbicides applied worldwide include glyphosate, paraquat, and diuron. Among these, the most well-known is glyphosate; since its introduction in the 1970s, its use spread quickly, facilitated cropping, but also implied in the growing appearance of resistant weeds, resulting from a natural process of plant adaptation, and decreasing its efficacy [111–113]. An important alternative to minimize this problem, in addition to integration with other control methods, would be diversification in the use of herbicides, including others with different mechanisms of action [114].

Few studies have investigated the compatibility between herbicides and inoculants. Madhaiyan et al. [106] evaluated the effects of different herbicides (butachlor, alachlor, atrazine, and 2,4-D) in liquid culture medium on the growth and metabolism of *G. diazotrophicus*. Cell growth was impaired only in the presence of 2,4-D, but all herbicides reduced nitrogenase activity, the production of IAA and GA$_3$, and the solubilization of P and Zn. The highest inhibition of nitrogenase activity (73.6% and 65.3%) was observed with alachlor and atrazine, respectively, while butachlor mostly affected production of IAA (53.3%), whereas 2.4-D mostly affected the production of GA$_3$ (78.8%). Butachlor was also responsible for the strongest reduction in the solubilization of P and Zn. Therefore, although herbicides do not affect cell survival, they significantly affect metabolism of *G. diazotrophicus* [106].

In an assay performed under greenhouse conditions, Angelini et al. [115] evaluated the effects of imazetapir, imazapic, S-metachlor, diclosulam, and glyphosate on diazotrophic bacteria in the soil during the cultivation of peanuts (*Arachis hypogaea* L.). The seeds were sown and the herbicide was sprayed on the soil surface. All herbicides caused reduction in cell concentration in both free and symbiotic diazotrophic bacteria, and this negative impact was confirmed under field conditions even one year after the application. Nitrogenase activity also reduced due to herbicides, except for glyphosate [115] (Table 2).

The ability of cyanobacteria to fix atmospheric nitrogen in flooded soils suitable for rice cultivation make this group an important ally in maintaining soil fertility and contributing to cereal yield. Thus, Dash et al. [116] evaluated the responses of cyanobacteria in rice plantation soil to the exposure of different agrochemicals, including the herbicide benthiocarb that was applied in one dose at the time of puddling. The herbicide decreased cell growth, which was even worse when combined with urea (used as a fertilizer). Benthiocarb reduced nitrogenase activity by between 13–27%, compared with the control without herbicide, and its combination with urea resulted in an even greater reduction in addition to a decrease in the nitrogen accumulation that reached 47% at 60 days.

**Table 2.** Effect of pesticides on the development of plants the seeds of which had been inoculated.

| Culture | Fungicide | Microorganism | local | Effect | Reference |
|---|---|---|---|---|---|
| Soybean (*Glycine max*) | Thiram [f] | *B. japonicum* | Greenhouse | Lower nodule number, nodule dry weight and nitrogenase activity | [21] |
| | Benomyl [f], Captan [f], Carbendazin [f], Carboxin [f], Difenoconazole [f], Thiabendazole [f], Thiram, Tolylfluanid [f] | *B. elkanii* (strain SEMIA 5019) + *B. japonicum* (strain SEMIA 5079) | Greenhouse and field | Reduction in the number of nodules and in the total N in grains | [24] |
| | Carbendazin [f] + Thiram [f]; Carboxin [f] + Thiram [f] | *B. elkanii* (strains 5019 + 587), *B. japonicum* (strain 5079) + *B. diazoefficiens* (strain SEMIA 5080) | Field | Reduction of nodulation efficiency. Reduction of N content and grain yield to SEMIA 587 with Carbendazin + Thiram | [101] |
| | Carbendazin [f] + Thiram [f] | *B. japonicum* (strain 5079) + *B. diazoefficiens* (strain SEMIA 5080) | Field | Reduction in the number of pods per plant and grains per plant | [102] |
| | Mefenoxam [f] + Fludioxonil [f] | *B. japonicum* | Field | Reduced grain yield and protein and oleic acid content | [103] |
| | Pyraclostrobin [f], thiophanato-methyl [f] and fipronil [i] | *B. japonicum* (strain 5079) and *B. diazoefficiens* (strain 5080) | Field | Less N accumulation in grains | [25] |
| Alfafa (*Medicago sativa*) | Methyl parathion [i], DDT [i] e pentachlorophenol [i] | *Sinorhizobium meliloti,* | Greenhouse | Reduction of nitrogenase activity, number of nodules and plant dry weight | [22] |
| Peanut (*Arachis hypogaea*). | Imazetapir [h], Imazapic [h], S-metachloro [h], Dichlosulam [h] and Glyphosate [h] | Diazotrophic bacteria present in the soil | Greenhouse | Reduction of cell concentration of free and symbiotic diazotrophic bacteria | [115] |
| | | | Field | Reduced cell concentration of free and symbiotic diazotrophic bacteria and reduced nitrogenase activity except for glyphosate | |
| Chickpeas (*Cicer arietinum* L.), pea (*Pisum sativum* L.), Mung beans (*Vigna radiata* L. Wiclzek) and lentil (*Lens esculentus*, = *Lens culinaris* Medik). | Pyriproxyfen [i] | Bacteria commonly present in the soil used | Pots in the field | Reduction in the number of nodules, in the dry weight of nodules, in the concentration of N in roots and in protein concentration in the grains | [23] |
| Rice (*Oryza sativa*) | *Benthicarb* [h] | *Cyanobacteria naturally found in the rice paddy soil* | Field | Decreased cell growth, nitrogenase activity and Naccumulation | [116] |
| Maize (*Zea mays*) | Pyraclostrobin [f], thyophanato-methyl [f] and fipronil [i] | *A. brasilense* (strains Ab-V5 and Ab-V6) | Greenhouse | Fewer branch roots, root hair and shorter root hair length | [26] |

[f] fungicide; [I] insecticide, [h] herbicide.



Concerning the symbiosis between legumes and rhizobia, in general, herbicides have been considered less toxic than fungicides and insecticides [100], with glyphosate being the one with lower toxicity [117,118]. Although the negative effects of glyphosate on *B. japonicum* growth were reported by King et al. [119], the doses in the experiment were far higher than those recommended for field application. With the release of genetically modified (GM) genotypes tolerant of herbicides, studies on the compatibility with the GM genotypes and herbicides have begun. In soybean, which represents the most used herbicide-tolerant species, glyphosate-resistant (Roundup Ready) pairs of nearly isogenic cultivars were evaluated in six field sites in Brazil for three crop seasons. Although the transgenic trait negatively affected some BNF variables, these effects had no significant impact on soybean grain yield, and no consistent differences between glyphosate and conventional herbicide application were observed on BNF-associated parameters [120]. Similar results were reported in 20 field trials performed with soybean with the *ahas* transgene, imidazolinone, and conventional herbicides [121]. However, it is worth mentioning that with the increasing doses of the herbicides, BNF can be reduced, especially under abiotic stressing conditions, as shown for glyphosate in soybean [122,123]. Interestingly, it has been long shown that several members of the family Rhizobiaceae were able to degrade glyphosate [124], ability that has been few explored, but that it can contribute to decrease the toxicity effects. Indeed, the search for indigenous and engineered microorganisms, used as isolated microbial species or microbial consortia can result in the complete mineralization of herbicides such as atrazine [125].

### 4.4. Compatibility with Mixtures (Fungicides, Insecticides, and Herbicides)

Approximately 70% of the pesticides available in the market contain mixtures of two or more types of fungicides and insecticides [126] and are often combined with herbicides at the time of application, aiming to facilitate the combined control of pests, diseases, and weeds. However, the damage to microbial inoculants increases with the number of combined chemicals. As previously mentioned, Standak® Top, one of the most used for treatments of seeds in several countries, especially for soybean, is composed of two fungicides and one insecticide; it affects the survival of *B. japonicum* and, especially, *B. elkanii* cells, with a drastic decrease verified after 7 days of contact [25]. It is worth mentioning that pre-inoculated soybean seeds have been in contact with Standak® Top for up to 90 days, often resulting in zero recoveries of rhizobial cells from seeds [126].

Another interesting observation in the study by Rodrigues et al. [25] was the changes in colony morphology, smaller with the increase of the exposure to the pesticide. However, regular colonies were recovered after the bacteria were grown under optimal conditions, indicating an adaptive mechanism to the stressful conditions when exposed to the pesticides.

Santos et al. [26] evaluated the compatibility of Standak® Top with *A. brasilense* strains Ab-V5 and Ab-V6. First, differences were observed between strains, with lower tolerance of Ab-V5, so that after 24 h of exposure the recovery of viable cells dropped from $4.56 \times 10^5$ to $4.37 \times 10^2$ cells seed$^{-1}$. In a greenhouse experiment with the combined strains, Standak® Top decreased the number of lateral roots and root hairs and resulted in shorter root hair length.

Pereira et al. [28] also reported the mortality of *A. brasilense* strains Ab-V5 and Ab-V6 in maize seeds treated with a mixture of fungicides and insecticides (metalaxyl-M + fludioxonil + thiamethoxam + abamectin). The cell survival rate after 12 h of inoculation of seeds treated with the pesticide was only 13.56%, whereas in untreated seeds it was 65.87%.

### 5. Are There Alternatives to the Challenges of Using Pesticides and Microbial Inoculants?

Remarkable advances in genetic engineering have occurred particularly in the last decade and are advancing towards obtaining plant genotypes resistant to abiotic stresses, pests and diseases [127–129]. However, large-scale agriculture to feed the increasingly growing population still demands the use of pesticides for at least the next two decades. Practical agriculture without pesticides may be a dream for most of society, but it is currently restricted to a few farmers, most in small properties, and there is no technology in the

research pipeline to make it feasible for the majority of the cropped areas in a short time. On the other hand, the use of bioproducts aimed at the total or partial replacement of chemicals used for the control of pests, diseases, weeds, and fertilizers is growing at a rate never seen before [17]. The major limitation, as we have shown in this review, is the low compatibility between pesticides and microbial inoculants applied to seeds. Therefore, there is an urgent need to develop alternatives to make microbial inoculants compatible with pesticides, and mainly to develop more biopesticides and other eco-friendly techniques to control pests and diseases. However, the compatibility of bioproducts must also be investigated, as not all microorganisms are compatible.

Ahemad and Khan [130] selected strains of *R. leguminosarum* that were tolerant to high concentrations of the insecticides fipronil and pyriproxifene used in peas. The tolerant strain MRP1 was characterized as the highest in the synthesis of IAA, siderophores, and exopolysaccharides (EPS); in a field trial performed in soil previously treated with insecticides, the strain contributed to the significant increases in nodulation, N and P contents of roots and shoots, and grain yield. The results show the feasibility of selecting strains tolerant of pesticides. One main limitation is the increasing number of pesticides used in agriculture, which would require multiple steps of microbial selection.

To develop pesticides less harmful to microbial bioinoculants, the inclusion of carriers for the active ingredients might be simpler than obtaining tolerant strains. Unfortunately, the pesticide industry has not demonstrated interest in following this strategy, probably because the chemical industry is financially more powerful than the inoculant industry and the limited interest in investing in less toxic molecules.

Another strategy could be to take advantage of microbial metabolites instead of living microbes [87]; however, this is only applicable to microorganisms that produce secondary metabolites useful to the host plants, such as *Azospirillum* sp. in grasses and pastures, and not by mechanisms that require living microorganisms, such as rhizobia to nodulate legumes.

Investment should be made in innovation in formulations, including cell protection to minimize or avoid the toxic effects of pesticides on microbial cells. The addition of protective molecules such as polymers, chemicals, or synthesized by microorganisms may help in this regard, for example, polyhydroxyburytate (PHB) [50,131,132] and biofilms [133].

In Brazil, with no short-term solutions for compatibility in sight, the only feasible strategy is physically avoiding the contact of inoculants with pesticides. Seed treatment with pesticides, for example, for soybeans that currently may contact up to 14 chemical products, an in-furrow application of biologicals has proven to be effective in guaranteeing the benefits to the crop. Despite the need for increasing the dose of inoculant applied in-furrow, the cost of inoculant is low in comparison with the benefits. A pioneering study confirming that the in-furrow inoculation of soybean with 2.5 times the concentration used for seed inoculation alleviated the effects of seed treatment with agrochemicals was published in 2010 [56]. Despite requiring that the farmers buy new equipment, 20% of the farmers in Brazil adopted this technique by 2020. Other strategies, with an emphasis on soil-spray and leaf-spray inoculation [57,86,134,135] have also been investigated and show some degree of effectiveness, but are not as effective as in-furrow and seed inoculations.

## 6. Final Remarks

The critical analysis of this review points to some certainties: (i) considering the technologies available today and those that should be available in coming years, large-scale agriculture to meet the increasing food demand will require pesticides; (ii) the demand for higher sustainability in agriculture, with bioproducts aiming at partially or fully replacing pesticides and fertilizers is increasing; (iii) inoculants or biofertilizers have been increasingly adopted by farmers, but their compatibility with pesticides, especially when used for seed treatment, is low; (iv) strategies to solve the incompatibility between pesticides and inoculants are needed, as those proposed until now are still very modest with regards to their feasibility; (v) the development of more biopesticides and other eco-friendly techniques to control pests and diseases and compatible with inoculants is needed.

Incompatibility between pesticides and inoculants affects cell survival and metabolism. The level of incompatibility with the pesticides depends on the active principle, formulation, doses, time of contact with the cells, and may vary with the bacterial species or strain. Despite the increasing market for biological products aiming at more sustainable agriculture [136], very few intellectual and economic investments have been made to search for compatibility of biological products with chemicals. Therefore, there is an urgent need to emphasize studies and development of innovative strategies to mitigate the incompatibility between pesticides and microbial inoculants.

**Author Contributions:** The writing, investigation and formal analysis were led by M.S.S. with support from T.F.R., M.H. and M.A.N. Regarding methodology and supervision, the authors contributed equally. M.H. led the validation and resources. All authors have read and agreed to the published version of the manuscript.

**Funding:** National Institute of Science and Technology, INCT-Plant-Growth Promoting Microorganisms for Agricultural Sustainability and Environmental Responsibility (CNPq 465133/2014-4, Fundação Araucária-STI 043/2019, CAPES) and Embrapa (20.19.02.009.00.01).

**Data Availability Statement:** All data and materials cited in the manuscript are freely available for the scientific community.

**Acknowledgments:** M.S.S. acknowledges a PhD fellowship from Araucaria Foundation of support to the Scientific and Technological Development of the State of Paraná. M.A. Nogueira and M. Hungria are research fellows of CNPq. We also thank the financial support given by the National Institute of Science and Technology, INCT-Plant-Growth Promoting Microorganisms for Agricultural Sustainability and Environmental Responsibility.

**Conflicts of Interest:** Authors declare no competing interest regarding the data or the manuscript.

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
