# Peer review of "The Challenge of Combining High Yields with Environmentally Friendly Bioproducts: A Review on the Compatibility of Pesticides with Microbial Inoculants"

_agronomy, doi:10.3390/agronomy11050870_

Round 1
Reviewer 1 Report
The review entitled “The challenge of combining high yields with environmentally-friendly bioproducts: A review on the compatibility of pesticides with microbial inoculants” covers an emerging topic in sustainable agriculture. The review adequately explores the most recent literature about the incompatibility between pesticides and bio-inoculants, although relevant works on this topic are still scarce. I do have a few suggestions/comments that authors could easily address and incorporate in the paper:
1) Add references for:
- lines 65-67 “In general, pesticides contain molecules that are potentially toxic to living cells. Depending on the specificity, pesticides can cause toxicity to cells of microorganisms, animals, and plants, often resulting in death after contact with the product.”
- lines 147-150 “In 1990, the average worldwide use of pesticides by cultivation area was OF 1.5 kg ha-1. Almost 30 years later, this value grew considerably, reaching an average of 2.63 kg ha-1 in 2018. The continents that applied more pesticides in 2018 were Asia and America, reaching 3.67 kg ha-1 and 3.52 kg ha-1, respectively.”
2) I suggest entitling chapter 2 “History and use of pesticides”. Similarly, chapter 4.1 “Compatibility of rhizobial inoculants with fungicides” since other microorganisms are not reported. In general, give less short and more powerful titles.
3) Maybe it would be useful to further sub-divided table 1 and table 2 into two sections: one for rhizobial inoculants and one for PGPB.
4) Finally, I would like that “the reported capacity of several rhizobia to degrade herbicides, such as glyphosates” to be further investigated, reporting in more detail the works 122-123, and, if possible, with other literature.
5) Moreover, the authors reported that “the mode of action of herbicides is generally related to cell division process, which inhibits a key enzyme/protein [110]” (lines 435-436). Please, explain in more detail the mode of action and specify the enzyme involved.
Author Response
Reviewer #1
Comments and Suggestions for Authors
The review entitled “The challenge of combining high yields with environmentally-friendly bioproducts: A review on the compatibility of pesticides with microbial inoculants” covers an emerging topic in sustainable agriculture. The review adequately explores the most recent literature about the incompatibility between pesticides and bio-inoculants, although relevant works on this topic are still scarce. I do have a few suggestions/comments that authors could easily address and incorporate in the paper:
Reply: Thanks for the suggestions that improved the review.
1) Add references for:
- lines 65-67 “In general, pesticides contain molecules that are potentially toxic to living cells. Depending on the specificity, pesticides can cause toxicity to cells of microorganisms, animals, and plants, often resulting in death after contact with the product.”
Reply: References added
- lines 147-150 “In 1990, the average worldwide use of pesticides by cultivation area was OF 1.5 kg ha-1. Almost 30 years later, this value grew considerably, reaching an average of 2.63 kg ha-1 in 2018. The continents that applied more pesticides in 2018 were Asia and America, reaching 3.67 kg ha-1 and 3.52 kg ha-1, respectively.”
Reply: Corrected.
2) I suggest entitling chapter 2 “History and use of pesticides”. Similarly, chapter 4.1 “Compatibility of rhizobial inoculants with fungicides” since other microorganisms are not reported. In general, give less short and more powerful titles.
Reply: Thanks for the suggestions. We changed the first title. In relation to the second, we also mentioned other microorganisms, and then we did not include rhizobia, but we improved the title. The title is now “Are pesticides and microbial inoculants compatible?”
3) Maybe it would be useful to further sub-divided table 1 and table 2 into two sections: one for rhizobial inoculants and one for PGPB.
Reply: Nowadays, the concept of mixed inoculants is exponentially growing. Mixture of microorganisms with different biological processes. Furthermore, microorganisms have been used with different purposes, e.g. there are rhizobia being used as PGPR in wheat in Argentina. Therefore, the concept that all these organisms are plant-growth promoters with different microbial processes is growing. Therefore, we ask not to split the Table.
4) Finally, I would like that “the reported capacity of several rhizobia to degrade herbicides, such as glyphosates” to be further investigated, reporting in more detail the works 122-123, and, if possible, with other literature.
Reply: We rephrased the sentence. And we added information about degradation ability, and added references.
5) Moreover, the authors reported that “the mode of action of herbicides is generally related to cell division process, which inhibits a key enzyme/protein [110]” (lines 435-436). Please, explain in more detail the mode of action and specify the enzyme involved.
Reply: We added examples about the mode of action and specificity.
Reviewer 2 Report
The manuscript deals with the compatibility between microbial biofertilizers and chemical pesticides. As authors document in the review we should minimize the accumulation of pollutants in agro-ecosystems. The use of microbial biofertilizers is a step toward the sustainable agriculture development and other step, even more important, is the substitution of chemical pesticides by alternative eco-friendly methods, including microbial biopesticides, increasingly available in the market.
When we choose to use microbial inoculants all the production system must be changed in order to work with all beneficial organisms and positive interactions in the agro-ecossystem. This wasn’t discussed in the paper as should have been.
Despite of the combined use of microbial fertilizers and pesticides is contradictory, due the impact of the chemicals in the living microbial inoculants, the higher dependence and use of pesticides in many world regions/continents, makes this review a relevant topic. However, there are some observations as follows:
Introduction:
L 60-63: “As such, the common scenario towards improving agricultural sustainability with feasible yields to guarantee food security includes the increasing use of bioproducts, such as microbial inoculants, together with pesticides, which are still indispensable for controlling pests and diseases”. This sentence needs a reference.
When farmers use microbial inoculants all the crop management must be adapted to the new products. The use of pesticides is not indispensable because there many other options for the crop protection. As for biofertilizers there are also many biopesticides, some of them also based in microorganisms, such as Tichoderma spp., Bacillus thurigiensis, Bacillus subtilis and other Bacillus spp., etc. If the use of pesticides were indispensable many systems like organic farming and other more sustainable farming methods, would be impossible, which is contradictory with the growing worldwide area devoted to these farming systems.
L69: “Usually, pesticides and inoculants are added together on the seeds”. This is not the common practice for the use of bioinoculants due the toxicity of the biopesticides, at least in many world regions. Please support this with a reference.
L72: “treating seeds with pesticides and inoculants several days before sowing, has become increasingly adopted by farmers”. Authors make a reference to a previous work from the team where no pesticides were added to the seeds. Please replace with a reference where pesticides and bioinoculants were added together.
L82: “The world will need more food, and to meet this increased demand, there is no doubt that pesticides should be employed”. The use of microbial inoculants requires alternative methods for pest and disease control, already available in the market, some of them based in microbial inoculants, too.
The authors shouldn’t be so taxative about the use of pesticides. Of course monocultures such as soyabean are problematic but more sustainable managmentent methods must be adopted by the farmers in order to benefit from the use of biofertilizers.
In L83-84 Authors claim that “there is also an increasing demand for environmentally-friendly inputs, including inoculants, to replace chemical fertilizers” but the same is aplied to chemical pesticides, which have higher impact both in human and environmental health and explain why many of them were or are being banned. Therefore the priority should be also to make compatible biofertilzers and biopesticides.
L72-L75: “However, crop management still requires the application of pesticides to achieve high yields to meet the world´s increasing demand for food. Given this scenario, the indications are the use of pesticides, but with a trend toward more environmentally-friendly formulations as the replacement by biological control”. See comment above.
L198: inoculant vehicles - inoculant carriers are more adequate
L221 – 1 x 10^9 viable cells. Please revise also in L509
L235 - Azospirillum brasilense – italicize the species name
L254 - Oryza sativa L. ; italicize A. brasilense (also other species along the text: ex: L376)
L260 – Please revise font size in the sentence and in the genera of bacteria along the text
L272-273 – Revise the sentence.
L521-530 – There are many regions of the world reducing the use of pesticides, and many alternatives are being developed, therefore the “heavy use of pesticides” as a requirement for to sustain crop productivity needs to be reformulated. There are contradictions in the text as commented before.
L529 – Authors claim that “there is an urgent need to develop alternatives to make microbial inoculants compatible with pesticides”. Due the impact of pesticides in microbial inoculants, there is an urgent need to develop more biopesticides and other eco-friendly techniques to control pests and diseases, some of them are already available in the market. This is not a dream but is the only way to protect ecossystems and human health. The same comment is applied to “Final remarks”
References:
Being a review, the reference list needs to be updated.
Please complete the DOI of the Ref 2: 10.7439/ijasr.v3i12.4410
Ref 11: Incomplete reference. Please don’t use an abstract from a scientific meeting instead of a good reference in a scientific journal because there are so many good papers on this subject.
Please complete Ref 41.
Author Response
Reviewer #2
Comments and Suggestions for Authors
The manuscript deals with the compatibility between microbial biofertilizers and chemical pesticides. As authors document in the review we should minimize the accumulation of pollutants in agro-ecosystems. The use of microbial biofertilizers is a step toward the sustainable agriculture development and other step, even more important, is the substitution of chemical pesticides by alternative eco-friendly methods, including microbial biopesticides, increasingly available in the market.
When we choose to use microbial inoculants all the production system must be changed in order to work with all beneficial organisms and positive interactions in the agro-ecossystem. This wasn’t discussed in the paper as should have been.
Despite of the combined use of microbial fertilizers and pesticides is contradictory, due the impact of the chemicals in the living microbial inoculants, the higher dependence and use of pesticides in many world regions/continents, makes this review a relevant topic. However, there are some observations as follows:
Reply: In this review, we had to choose topics and we decided to emphasize the challenge of compatibility. This because the use of inoculant has not required the change of the production system. One major example is the soybean in Brazil, where 80% of the cropped are is inoculated, but all the other chemical products are also used.
Thanks for considering our review a relevant topic, and we thank also to the suggestions.
Introduction:
1) L 60-63: “As such, the common scenario towards improving agricultural sustainability with feasible yields to guarantee food security includes the increasing use of bioproducts, such as microbial inoculants, together with pesticides, which are still indispensable for controlling pests and diseases”. This sentence needs a reference.
Reply: References included
2) When farmers use microbial inoculants all the crop management must be adapted to the new products. The use of pesticides is not indispensable because there many other options for the crop protection. As for biofertilizers there are also many biopesticides, some of them also based in microorganisms, such as Tichoderma spp., Bacillus thurigiensis, Bacillus subtilis and other Bacillus spp., etc. If the use of pesticides were indispensable many systems like organic farming and other more sustainable farming methods, would be impossible, which is contradictory with the growing worldwide area devoted to these farming systems.
3) L69: “Usually, pesticides and inoculants are added together on the seeds”. This is not the common practice for the use of bioinoculants due the toxicity of the biopesticides, at least in many world regions. Please support this with a reference.
Reply: Step by step the biological are replacing the chemicals. However, today, at least for major crops, we have a mixture of chemical and biological products. As we mentioned in the first paragraph, one major example is soybean inoculation in South America, together with an astonishing amount of chemicals.
Reply: We added references.
4) L72: “treating seeds with pesticides and inoculants several days before sowing, has become increasingly adopted by farmers”. Authors make a reference to a previous work from the team where no pesticides were added to the seeds. Please replace with a reference where pesticides and bioinoculants were added together.
Reply: Sorry for the mistake. We re-phrased and added the correct references for this statement.
5) L82: “The world will need more food, and to meet this increased demand, there is no doubt that pesticides should be employed”. The use of microbial inoculants requires alternative methods for pest and disease control, already available in the market, some of them based in microbial inoculants, too.
The authors shouldn’t be so taxative about the use of pesticides. Of course monocultures such as soyabean are problematic but more sustainable managmentent methods must be adopted by the farmers in order to benefit from the use of biofertilizers.
Reply: Unfortunately we still should have some decades ahead before we can have world large-scale production without pesticides. Large-scale production of few crops are increasing, and this also increases the incidence of pests and diseases. But we changed the phrase, indicating that at least in a middle term period they should continue to be required.
6) In L83-84 Authors claim that “there is also an increasing demand for environmentally-friendly inputs, including inoculants, to replace chemical fertilizers” but the same is aplied to chemical pesticides, which have higher impact both in human and environmental health and explain why many of them were or are being banned. Therefore the priority should be also to make compatible biofertilzers and biopesticides.
Reply: Yes, correct. We included in the phrase the biological and chemical pesticides.
7) L72-L75: “However, crop management still requires the application of pesticides to achieve high yields to meet the world´s increasing demand for food. Given this scenario, the indications are the use of pesticides, but with a trend toward more environmentally-friendly formulations as the replacement by biological control”. See comment above.
Reply: As we explained before, we re-phrased the sentence.
8) L198: inoculant vehicles - inoculant carriers are more adequate
Reply: Changed
9) L221 – 1 x 10^9 viable cells. Please revise also in L509
Reply: In the first case we changed to CFU (colony forming units). The other times where we cite viable cells is because the methodology defined by the reference 27 mentions viable cells.
10) L235 - Azospirillum brasilense – italicize the species name
Reply: I think that it was a problem in the transformation of the text, as we checked names in our word text and they were all in italic. But we checked everything.
11) L254 - Oryza sativa L. ; italicize A. brasilense (also other species along the text: ex: L376)
Reply: Please see question #10. Sorry, we believe that there was a problem in the transformation of the text sent to you.
12) L260 – Please revise font size in the sentence and in the genera of bacteria along the text.
Reply: We revised everything. Again, we think that there was a problem in the transformation of the text.
13) L272-273 – Revise the sentence.
Reply: In our manuscript the sentence was “With important properties and taking advantage of different microbial processes, the inoculation with mixes of bacteria has gained increased attention”. We have changed to “Therefore, the inoculation with mixes of bacteria has gained increased attention for taking advantage of microbes with different properties”
14) L521-530 – There are many regions of the world reducing the use of pesticides, and many alternatives are being developed, therefore the “heavy use of pesticides” as a requirement for to sustain crop productivity needs to be reformulated. There are contradictions in the text as commented before.
Reply: We deleted the heavy, and with the complementation that at least for the next two decades, we believe that it now represents the right picture.
15) L529 – Authors claim that “there is an urgent need to develop alternatives to make microbial inoculants compatible with pesticides”. Due the impact of pesticides in microbial inoculants, there is an urgent need to develop more biopesticides and other eco-friendly techniques to control pests and diseases, some of them are already available in the market. This is not a dream but is the only way to protect ecossystems and human health. The same comment is applied to “Final remarks”
Reply: We added the comments and we do believe that they enrich the both discussion and the final remarks. References:
16) Being a review, the reference list needs to be updated.
Reply: We have a relatively long list of references that we tried to be composed by the most relevant, most cited, most important papers, as well as those unique for same points raised in the review.
17) Please complete the DOI of the Ref 2: 10.7439/ijasr.v3i12.4410
Reply: Sorry for the mistake, it was corrected.
18) Ref 11: Incomplete reference. Please don’t use an abstract from a scientific meeting instead of a good reference in a scientific journal because there are so many good papers on this subject.
Reply: The reference was because we were referring to the “micro green revolution” in this reference. We completed the reference.
19) Please complete Ref 41.
Reply: Reference completed.